# O-GlcNAcylation enhances CPS1 catalytic efficiency for ammonia and promotes ureagenesis

Leandro R. Soria [1] ✉, Georgios Makris[2], Alfonso M. D'Alessio[1],
Angela De Angelis[1], Iolanda Boffa[1], Veronica M. Pravata [3],
Véronique Rüfenacht[2], Sergio Attanasio[1], Edoardo Nusco[1], Paola Arena [1],
Andrew T. Ferenbach[3], Debora Paris[4], Paola Cuomo[4], Andrea Motta [4],
Matthew Nitzahn[5], Gerald S. Lipshutz [5,6], Ainhoa Martínez-Pizarro[7],
Eva Richard [7], Lourdes R. Desviat [7], Johannes Häberle [2],
Daan M. F. van Aalten [3] & Nicola Brunetti-Pierri [1,8,9] ✉

Life-threatening hyperammonemia occurs in both inherited and acquired liver diseases affecting ureagenesis, the main pathway for detoxification of neurotoxic ammonia in mammals. Protein O-GlcNAcylation is a reversible and nutrient-sensitive post-translational modification using as substrate UDP-GlcNAc, the end-product of hexosamine biosynthesis pathway. Here we show that increased liver UDP-GlcNAc during hyperammonemia increases protein O-GlcNAcylation and enhances ureagenesis. Mechanistically, O-GlcNAcylation on specific threonine residues increased the catalytic efficiency for ammonia of carbamoyl phosphate synthetase 1 (CPS1), the rate-limiting enzyme in ureagenesis. Pharmacological inhibition of O-GlcNAcase, the enzyme removing O-GlcNAc from proteins, resulted in clinically relevant reductions of systemic ammonia in both genetic (hypomorphic mouse model of propionic acidemia) and acquired (thioacetamide-induced acute liver failure) mouse models of liver diseases. In conclusion, by fine-tuned control of ammonia entry into ureagenesis, hepatic O-GlcNAcylation of CPS1 increases ammonia detoxification and is a novel target for therapy of hyperammonemia in both genetic and acquired diseases.

Ammonia is continuously generated from the breakdown of proteins and other nitrogen-containing molecules, and from the gut microbiome. Under physiological conditions, plasma ammonia concentrations are maintained below of 50–75 μM in neonates, and 40–50 μM in adults[1]. Free ammonia crosses the blood–brain barrier and is neurotoxic and potentially lethal at high concentrations[2]. Hence, to prevent hyperammonemia, waste nitrogen must be converted into urea and glutamine by the liver[3]. However, in patients with inherited or acquired diseases, such as urea cycle disorders (UCD), organic acidemias, or acute and chronic liver diseases, ammonia is poorly removed from the circulation resulting in life-threatening hyperammonemia. UCD and organic acidemias are the most common genetic disorders presenting with hyperammonemia, and their cumulative incidences is up to 1:35,000 and 1:20,000 births, respectively[1]. The extent of irreversible neurologic damage depends on the brain maturational stage and on the magnitude and duration of the hyperammonemia. The developing brain is especially susceptible to hyperammonemia that leads to severe cognitive impairment, seizures and cerebral palsy[4]. Damage may become irreversible in case of prolonged hyperammonemia or when blood ammonia reaches levels between 200 and 500 μM in the first years of life[1]. In

adults hyperammonemia leads to hepatic encephalopathy characterized by altered mental status and coma. Independently from the causes, hyperammonemia is life-threatening and requires immediate and thorough treatment. Despite available therapies, hyperammonemia remains an extremely challenging medical condition with high morbidity and mortality, and survival closely correlates with blood ammonia concentrations. Therefore, several treatments have been developed or are under investigation to lower toxic ammonia[5].

O-linked β-N-acetylglucosamination (O-GlcNAcylation) is a fast-cycling and nutrient-sensitive post-translational modification (PTM) targeting serine (Ser) and threonine (Thr) residues of cytoplasmic, mitochondrial, and nuclear proteins[6]. In response to physiologic and environmental cues, this PTM coordinates various cellular processes including gene expression, signal transduction, cell cycle, and metabolism[7]. More than 6000 proteins are known to undergo O-GlcNAcylation[8,9]. The substrate of O-GlcNAcylation is uridine diphosphate-GlcNAc (UDP-GlcNAc), the end-product of the hexosamine biosynthesis pathway (HBP), which integrates metabolic flux through several pathways linked to intake of nutrients, including carbohydrates (glucose), amino acids (glutamine), fatty acids (acetyl-CoA) and nucleotides (uridine)[10]. O-GlcNAcylation is operated by O-GlcNAc transferase (OGT) that covalently attaches GlcNAc to proteins, whereas O-GlcNAcase (OGA) removes the O-GlcNAc. O-GlcNAc can be installed and removed on Ser and Thr residues multiple times during the lifespan of a protein affecting its folding, stoichiometry, interaction, activity, degradation, and sub-cellular localization[11]. In the liver, O-GlcNAcylation has been involved in insulin signaling, gluconeogenesis, lipogenesis, bile acid synthesis, circadian clock, and autophagy[11]. Moreover, protein O-GlcNAcylation increases in response to several stress conditions, acting as an adaptive process and promoting cell protection[12]. In the present study, we unravel the role of hepatic O-GlcNAcylation in ureagenesis and ammonia detoxification.

## Results

### Hyperammonemia increases hepatic O-GlcNAcylation
As a metabolic sensor, O-GlcNAcylation is responsive to glutamine and uridine that are incorporated into UDP-GlcNAc (Fig. 1a). Elevated protein O-GlcNAcylation was previously found in rat astrocytes incubated with high concentrations of ammonia[13]. To investigate hepatic O-GlcNAcylation in hyperammonemia, we first evaluated liver UDP-GlcNAc and the amounts of O-GlcNAc-processing enzymes in a mouse model of acute hyperammonemia[14], in which acute and transient elevation of blood ammonia was induced in C57BL/6 wild-type (WT) mice by intraperitoneal (i.p.) injections of 10 mmol/kg of ammonium chloride. The livers of mice with acute hyperammonemia showed increased concentrations of glutamine, uridine and UDP-GlcNAc (Fig. 1b), while both protein amount (Fig. 1c, d) and catalytic activity (Supplementary Fig. 1a) of glutamine-fructose-6-phosphate aminotransferase 1 (GFPT1), the rate-limiting enzyme in HBP, were unaffected. Moreover, liver OGT and OGA protein amounts were unchanged during hyperammonemia (Fig. 1c, d). Western blots [by two different antibodies RL2 (Fig. 1e) and CTD110.6 (Supplementary Fig. 1b) specific for O-GlcNAc-modified proteins] and immunohistochemistry (Fig. 1f) of livers harvested at various time points after nitrogen load showed increased O-GlcNAc-modified proteins in hepatocytes of mice injected with ammonium chloride compared to control mice injected with sodium chloride. In ammonium-treated mice, increased protein O-GlcNAcylation was detected in mitochondrial and nuclear subcellular fractions, whereas the increase of O-GlcNAc-modified proteins was less prominent in the cytosolic fraction (Supplementary Fig. 1c). Taken together, these results support a robust substrate-mediated increase of hepatic protein O-GlcNAcylation induced by hyperammonemia.

### Hepatic protein O-GlcNAcylation regulates ureagenesis
To interrogate the role of hepatic protein O-GlcNAcylation in ureagenesis, we evaluated the incorporation into urea of either [15]N or [13]C from [15]N-ammonium chloride or [13]C-sodium acetate respectively, in mice with liver overexpression of OGT under the control of the hybrid liver-specific promoter (HLP) mediated by an hepatotropic adeno-associated viral (AAV) vector. Compared to control mice injected with the same AAV vector serotype expressing the green fluorescent protein (GFP) under the control of the same HLP promoter, WT mice injected intravenously (i.v.) with an AAV-vector delivering a Myc-tagged murine OGT showed increased levels of O-GlcNAc-modified proteins with no changes in OGA protein levels (Fig. 2a–c). Moreover, enhancement of protein O-GlcNAcylation was associated with increased ureagenesis evaluated by [15]N-urea (Fig. 2d) and [13]C-urea (Supplementary Fig. 2a). Increased ureagenesis was not dependent on augmented expression of urea cycle enzymes, which remained unaffected in AAV–OGT injected mice compared to controls (Supplementary Fig. 2b, c). Next, we investigated the consequence of impaired protein O-GlcNAcylation after pharmacological inhibition of OGT. The OGT inhibitor (ST045849, 40 mg/kg i.p.) reduced hepatic O-GlcNAc-modified proteins (Supplementary Fig. 2d) and [15]N-ureagenesis (Supplementary Fig. 2e). However, reduction in hepatic OGT activity also resulted in liver injury, as shown by elevated serum alanine transaminase (ALT) (Supplementary Fig. 2f), consistent with the previous findings of hepatotoxicity in liver-specific OGT knock-out mice[15]. Because liver damage impairs ureagenesis and ammonia clearance capacity, consequences of OGT inhibition on ureagenesis could not be directly investigated. To overcome this limitation, we reduced hepatic protein O-GlcNAcylation in WT mice by i.v. injections of an AAV vector expressing a Myc-tagged murine OGA under the control of the liver-specific HLP promoter (AAV-OGA). Hepatic OGA overexpression did not affect OGT protein levels (Fig. 2e, f) but reduced protein O-GlcNAcylation (Fig. 2g) and resulted in higher blood ammonia after ammonium chloride challenge and reduced incorporation of [15]N into urea compared with AAV-GFP-injected mice (Fig. 2h, i). Notably, serum ALT activities were not increased by hepatic OGA overexpression (Supplementary Fig. 2g), thus excluding liver damage. Together, these data suggest a direct correlation between hepatic O-GlcNAcylation and ureagenesis.

### Downregulation or inhibition of OGA enhances ureagenesis and ammonia detoxification
Hepatic downregulation of OGA by i.v. injections of a siRNA against murine OGA RNA into WT mice resulted in increased protein O-GlcNAcylation (Fig. 3a, b), reduced concentrations of blood ammonia (Fig. 3c), and increased blood [15]N-urea (Fig. 3d) after ammonium chloride challenge, compared to mice injected with a non-targeting negative control siRNA. Improved ammonia clearance was not dependent on increased expression of urea cycle enzymes in mice treated with siRNA against OGA compared to controls, as shown by Western blotting for urea cycle enzymes (Supplementary Fig. 3a, b). Ammonia detoxification was investigated in WT mice with acute hyperammonemia by treatment with Thiamet-G (40 mg/kg, i.p. for 2 days), a selective and potent OGA inhibitor[16]. Despite reduced OGT expression that is consistent with previous studies[17], Thiamet-G resulted in increased hepatic protein O-GlcNAcylation (Supplementary Fig. 3c) and [15]N-urea production (Fig. 3e), and clinically relevant reductions in blood ammonia concentrations (Fig. 3f), without affecting protein amounts of urea cycle enzymes (Supplementary Fig. 3d, e). Inhibition of OGA by PUGNAc (7 mg/kg i.p. for 2 days), another OGA inhibitor, also increased liver protein O-GlcNAcylation, [15]N-ureagenesis and protected against acute ammonia overload (Supplementary Fig. 4a–c). Neither siRNA-mediated nor pharmacological inhibition of OGA affected blood concentrations of glutamine or liver amount of glutamine synthetase, the additional waste

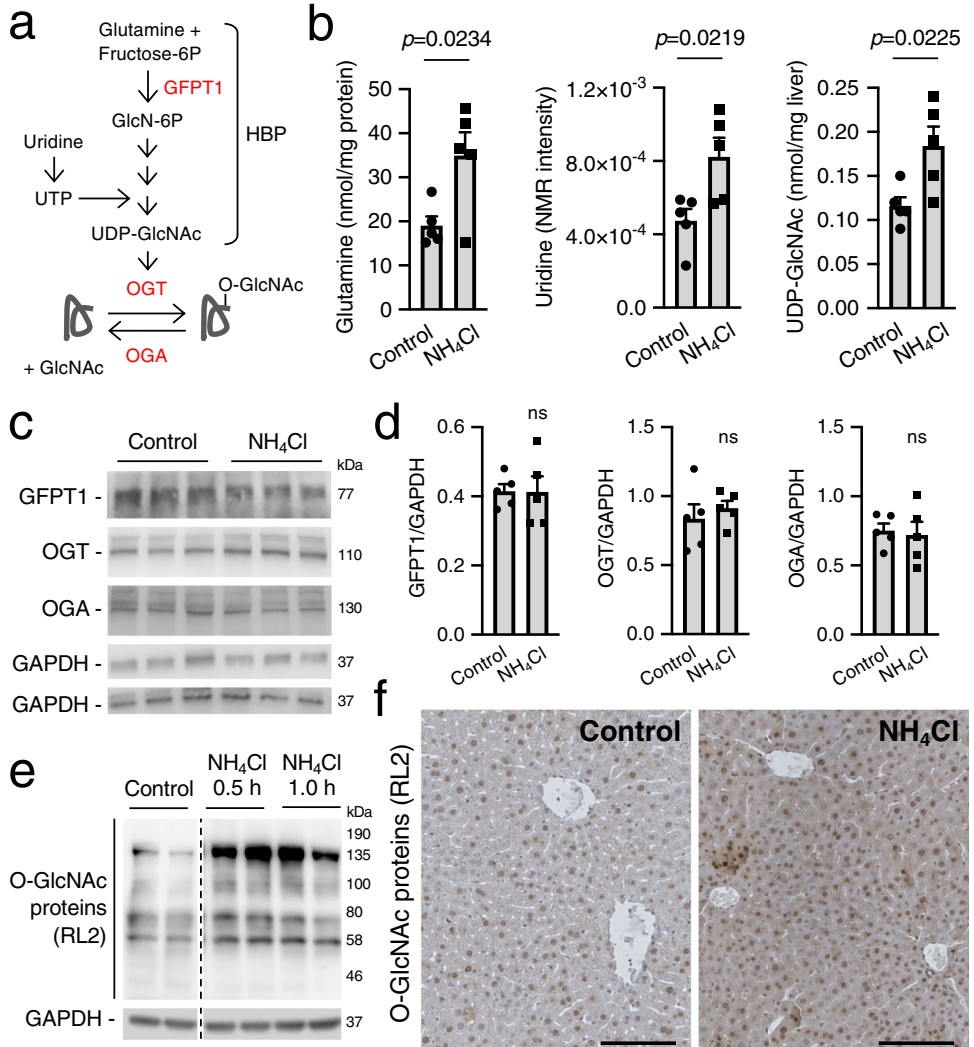

**Fig. 1 | Hyperammonemia enhances hepatic O-GlcNAcylation. a** Schematic depiction of the HBP and O-GlcNAcylation. Glutamine and uridine feed into the HBP that produces UDP-GlcNAc, while GFPT1 is the rate-limiting step of UDP-GlcNAc synthesis. Protein O-GlcNAcylation is operated by OGT while OGA removes the O-GlcNAc from proteins. **b** Hepatic content of glutamine, uridine and UDP-GlcNAc in livers of C57BL/6 wild-type (WT) mice at 1.0 h after the intraperitoneal (i.p.) injection of ammonium chloride (NH₄Cl) (10 mmol/kg) compared to livers of mice that received i.p. injection of sodium chloride (Control) (n = 5 mice/group). p = 0.0234, p = 0.0219, p = 0.0225. (Unpaired t-test). **c, d** Western blots and densitometric quantifications of GFPT1, OGT and OGA in livers of WT mice harvested 1.0 h after the i.p. injection of either sodium chloride (Control) or NH₄Cl (10 mmol/kg) (n = 5 mice/group). **e** Western blot for O-GlcNAc proteins with RL2 antibody in livers of WT mice harvested at 0.5 and 1.0 h after i.p. injections of NH₄Cl (10 mmol/kg) compared to sodium chloride-injected mice. Samples were run on the same gel but were non-contiguous. **f** Representative immunohistochemistry images for O-GlcNAc proteins in livers of WT mice harvested 0.5 h after the i.p. injection of either sodium chloride (Control) or NH₄Cl (10 mmol/kg). Scale bars: 100 μm. GAPDH was used as loading control. All values are shown as averages ± SEM. ns, no statistically significant difference. Experiments in panels **e** and **f** were performed twice.

nitrogen removal system in mammals (Supplementary Fig. 5a–d). Taken together, these results suggest that OGA is a *druggable* candidate for therapy of hyperammonemia.

## O-GlcNAcylation of CPS1 improves enzyme kinetics for ammonia

Because O-GlcNAc-mediated enhancement of ureagenesis was not dependent on increased expression of any of the urea cycle enzymes, and protein O-GlcNAcylation is known to modulate enzyme activity[10,11] among other functions, we hypothesized that O-GlcNAcylation regulates the enzyme activity of one or more urea cycle enzymes. Immunoprecipitation of mouse liver lysates with an anti-O-GlcNAc antibody followed by mass spectrometry analysis revealed carbamoyl phosphate synthetase 1 (CPS1) as the only enzyme involved in urea synthesis modified by O-GlcNAc (Supplementary Fig. 6a and Supplementary Data 1). Consistently, CPS1 undergoes O-GlcNAcylation according to two

comprehensive databases of O-GlcNAcylated proteins[8,9]. CPS1 is the first and rate-limiting step of ureagenesis that catalyzes the direct incorporation of ammonia into urea cycle intermediates in mitochondria. Notably, inherited CPS1 deficiency is one of the most severe UCD[18]. Dynamic O-GlcNAcylation of CPS1 was confirmed by immunoprecipitation of CPS1 in livers of WT mice injected i.p. with vehicle or Thiamet-G (Fig. 4a, b). Pre-incubation of liver lysates with the bacterial OGA orthologue from *Clostridium perfringens* (CpOGA) that removes O-GlcNAc from proteins[19] resulted in loss of CPS1 immunoreactivity by the anti-O-GlcNAc antibody (Fig. 4a and Supplementary Fig. 6b). Moreover, AAV-mediated hepatocyte-specific delivery of OGT or OGA resulted in increased or decreased O-GlcNAcylation of CPS1, respectively (Supplementary Fig. 6c). Notably, O-GlcNAcylation of CPS1 was also observed in human hepatocytes (Fig. 4c). O-GlcNAc modification may affect either protein stability and/or activity[10,11] but the amount of CPS1 in isolated mitochondria (Supplementary Fig. 6d) and CPS1 catalytic

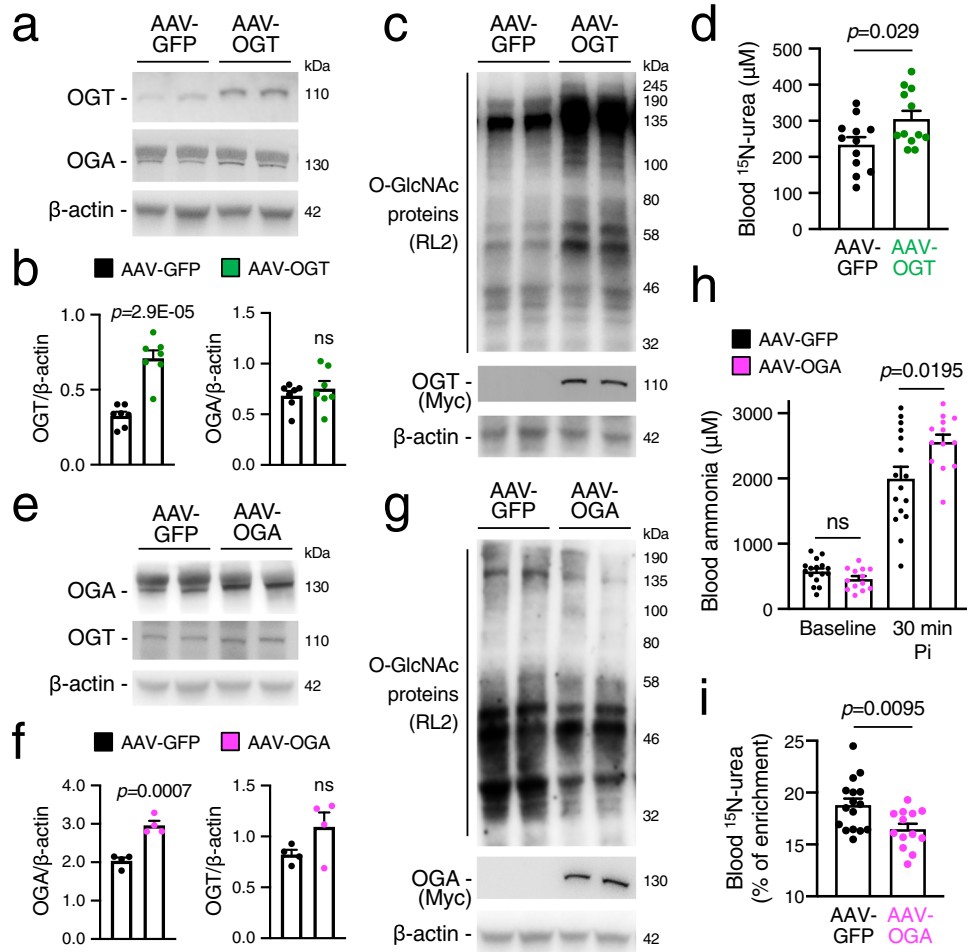

**Fig. 2 | Hepatic protein O-GlcNAcylation regulates ureagenesis. a** Western blots for OGT and OGA of livers of C57BL/6 wild-type (WT) mice harvested four weeks after the intravenous (i.v.) injections of $1 \times 10^{13}$ genome copies (GC)/kg of AAV-OGT vector expressing the OGT under the control of a liver-specific promoter or an AAV-GFP expressing the GFP reporter gene under the control of the same promoter, as controls. **b** Densitometric quantifications ($n = 7$ mice/group). $p = 2.9E-05$ (Unpaired $t$-test). **c** Western blot for O-GlcNAc proteins and Myc-tagged OGT in livers of AAV-OGT- and AAV-GFP-injected WT mice. **d** $^{15}N$-urea in blood 30 min after i.p. injections of $^{15}NH_4Cl$ (2 mmol/kg) in AAV-OGT- and AAV-GFP-injected WT mice ($n = 12$ mice/group). $p = 0.029$ (Unpaired $t$-test). **e** Hepatic OGA and OGT proteins by Western blot in WT mice injected i.v. with AAV-OGA vector ($1 \times 10^{13}$ GC/kg) at four weeks post-vector injection. Control mice were injected with the same dose of AAV-GFP. **f** Densitometric quantifications ($n = 4$ mice/group). $p = 0.0007$ (Unpaired $t$-test). **g** Western blot for O-GlcNAc proteins and Myc-tagged OGA in livers of WT mice injected with AAV-OGA or AAV-GFP at four weeks post-vector administration. **h** Blood ammonia in WT mice injected with AAV-OGA ($n = 13$ mice/group) or AAV-GFP-treated ($n = 16$ mice/group) at baseline and 30 min after i.p. injection of $NH_4Cl$ (10 mmol/kg). $p = 0.0195$ (Unpaired $t$-test). **i** Isotopic enrichment of $^{15}N$-urea in blood 30 min after i.p. injection of $^{15}NH_4Cl$ (10 mmol/kg) in WT mice injected with AAV-OGA ($n = 13$) or AAV-GFP ($n = 16$). $p = 0.0095$ (Unpaired $t$-test). β-actin was used as loading control. All values are shown as averages ± SEM. Pi, post-injection; ns, no statistically significant difference.

activity were both unaffected (Supplementary Fig. 6e) by Thiamet-G. However, protein O-GlcNAcylation can also modulate the kinetic efficiency (specificity constant: $k_{cat}/K_M$) of enzymes towards their substrates[20–22]. Consistently, increased O-GlcNAcylation of CPS1 by Thiamet-G augmented CPS1 kinetic efficiency towards ammonia (Fig. 4d), whereas kinetic efficiency was unaffected by the other CPS1 substrates (bicarbonate and ATP) or by its allosteric activator *N*-acetyl-glutamate (NAG) (Supplementary Fig. 6f) in liver lysates of mice treated with Thiamet-G compared to vehicle-treated controls. Notably, the improved kinetic efficiency for ammonia induced by Thiamet-G was blunted by pre-incubation of liver lysates with *Cp*OGA, confirming that the increased kinetic efficiency is mediated by O-GlcNAcylation (Fig. 4e). In addition, we ruled out that increased ureagenesis induced by Thiamet-G was dependent on CPS1 acetylation, a PTM operated by mitochondrial sirtuin 5 (SIRT5), that was previously found to regulate enzyme activity and ureagenesis[23–25]. By Western blot, protein levels of SIRT5 and acetylated CPS1 were indeed unchanged in livers of WT mice treated with Thiamet-G compared to controls (Supplementary

Fig. 7a, b). Moreover, protein–protein interaction between SIRT5 and CPS1 was unaffected (Supplementary Fig. 7c).

To identify the site(s) of O-GlcNAcylation on CPS1, we performed high-resolution electron transfer dissociation mass spectrometry (ETD/MS) after immunoprecipitation of CPS1 from Thiamet-G-treated livers. We found that CPS1 is modified with O-GlcNAc at Thr109 and Thr110 located on protein N-terminal region and at Thr1078 located in the carbamate phosphorylation domain (Fig. 4f and Supplementary Fig. 8a–c). In addition to O-GlcNAcylation, Thr109 was also found to be phosphorylated whereas no phosphorylation was detected on Thr110 and Thr1078 (Supplementary Data 2). All three Thr residues are conserved between mouse and human CPS1 (Fig. 4g) and located in disordered regions (Fig. 4h)[26], favoring a dynamic access of the O-GlcNAc-cycling enzymes. To investigate the functional relevance of these residues, we measured the catalytic activity of CPS1 enzymes with each of the Thr residues mutated into alanine (Ala) as singlet or triplet mutants. Although O-GlcNAcylation was detected in baculovirus/insect cells, overexpressed CPS1 was lacking O-GlcNAcylation in these

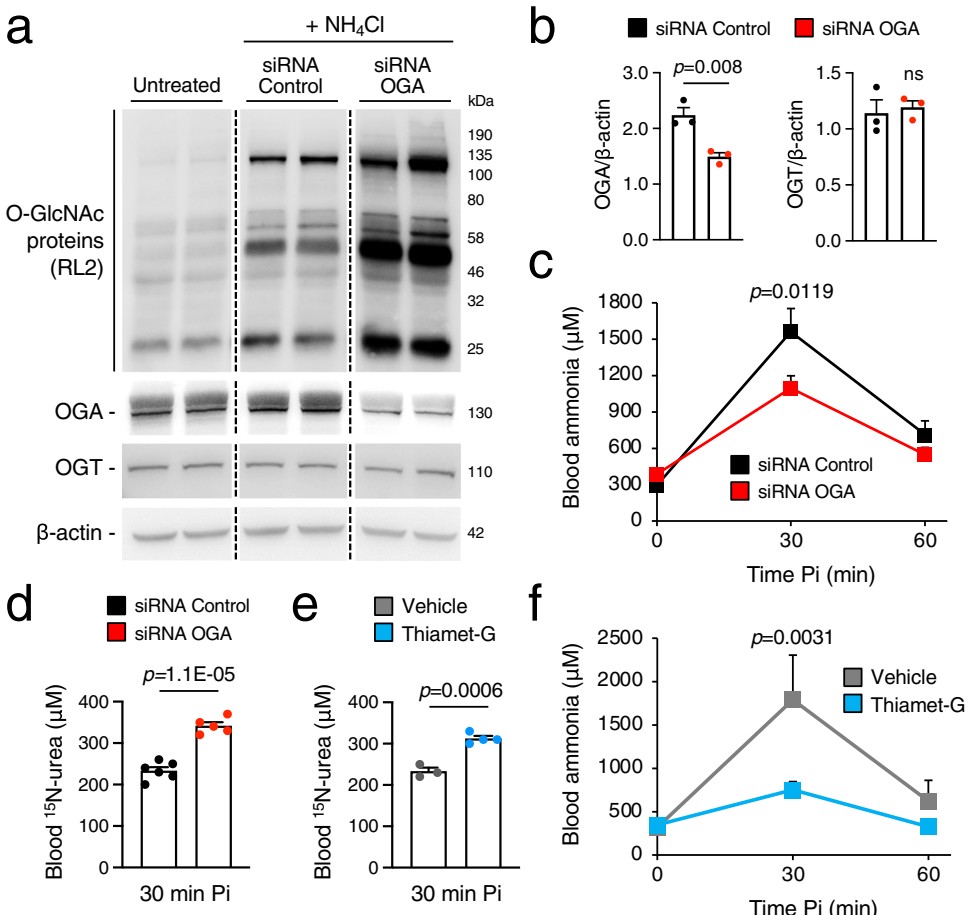

**Fig. 3 | Downregulation or pharmacologic inhibition of OGA enhances urea-genesis and ammonia detoxification. a** Western blot analyses of O-GlcNAc proteins, OGA and OGT proteins in livers harvested 1.0 h after the i.p. injections of $NH_4Cl$ (10 mmol/kg) in C57BL/6 wild-type (WT) mice injected i.v. with siRNA (1 mg/kg) against murine OGA RNA or with a non-targeting negative control siRNA 6 days before the ammonia challenge. Control samples of untreated mouse livers are also shown. Samples were run on the same gel but were non-contiguous. **b** Densitometric quantifications of the OGA and OGT Western blots ($n = 3$ mice/group). $p = 0.0081$ (Unpaired $t$-test). **c** Blood ammonia at baseline and after i.p. injection of $NH_4Cl$ (10 mmol/kg) in negative control siRNA ($n = 10$) and OGA-siRNA-treated ($n = 12$) mice.

$p = 0.0119$ (Two-way ANOVA). **d** $^{15}$N-urea in blood 30 min after i.p. injection of $^{15}NH_4Cl$ tracer (10 mmol/kg) in negative control siRNA ($n = 6$ mice/group) and OGA-siRNA-treated ($n = 5$ mice/group) WT mice. $p = 1.1E{-}05$ (Unpaired $t$-test). **e** $^{15}$N-urea in blood 30 min after i.p. injection of $^{15}NH_4Cl$ tracer (10 mmol/kg) in WT mice injected i.p. with Thiamet-G (40 mg/kg for two days) or vehicle (PBS) ($n = 3–4$ mice/group). $p = 0.0006$ (Unpaired $t$-test). **f** Blood ammonia at baseline and after i.p. injections of $NH_4Cl$ (10 mmol/kg) in WT mice injected i.p. with Thiamet-G (40 mg/kg for two days) ($n = 7$) or vehicle ($n = 5$). $p = 0.0031$ (two-way ANOVA). β-actin was used as loading control. All values are shown as averages ± SEM. Pi post-injection, ns no statistically significant difference.

cells (Supplementary Fig. 9a, b). When each of these Thr residues are mutagenized into Ala, CPS1 enzyme activity was significantly reduced, supporting the importance of these residues for enzyme function (Fig. 4i). While the singlet mutant enzymes showed protein amounts comparable to WT, the triplet had mild reduction in protein abundance (Fig. 4j). Moreover, the triple mutant enzyme transfected into Huh7 cells resulted in reduced ammonia removal from media compared to control cells transfected with the WT CPS1 (Supplementary Fig. 9c, d). In summary, these data suggest that O-GlcNAcylation regulates ureagenesis by modulation of CPS1 function.

**OGA inhibition reduces hyperammonemia in mouse models of both inherited and acquired liver diseases**

Secondary hyperammonemia occurs during organic acidemias as a consequence of the accumulation of toxic substrates that impair CPS1 activity. Hence, in the hypomorphic mouse model of propionyl-CoA carboxylase subunit alpha (PCCA) deficiency[27], we investigated the efficacy of OGA inhibition for therapy of hyperammonemia. In $Pcca^{-/-}$ (A138T) mice, i.p. administration of Thiamet-G reduced blood ammonia compared to vehicle-treated mice (Fig. 5a). Consistent with the human disease, $Pcca^{-/-}$ (A138T) mice have an elevated propionylcarnitine (C3)/

acetylcarnitine (C2) ratio in blood[27], that was found to be unaffected by the Thiamet-G (Fig. 5b), suggesting that the reduced blood ammonia concentration is not a direct effect of Thiamet-G on propionyl-CoA carboxylase residual activity. Compared to WT, $Pcca^{-/-}$ (A138T) mice showed similar amounts of O-GlcNAc-modified proteins, that increased after Thiamet-G treatment (Supplementary Fig. 10a). Increased O-GlcNAcylation of CPS1 was confirmed in livers of $Pcca^{-/-}$ (A138T) mice injected with Thiamet-G (Fig. 5c). Finally, we investigated the efficacy of Thiamet-G for therapy of hyperammonemia induced by acute liver failure, a life-threatening condition associated with impaired ammonia detoxification capacity. Liver failure in WT mice was induced by i.p. administration of thioacetamide (TAA), resulting in hyperammonemia[14]. A mild reduction of protein O-GlcNAcylation was detected in livers of TAA-treated mice, but Thiamet-G treatment was still effective at increasing protein O-GlcNAcylation in livers of TAA-treated mice (Supplementary Fig. 10b). Notably, compared to vehicle-treated controls, mice treated with Thiamet-G showed decreased blood ammonia during acute liver failure (Fig. 5d) despite similar levels of serum ALT observed between the two groups (Fig. 5e) and increased O-GlcNAcylation of CPS1 (Fig. 5f), thus suggesting that reduction of blood ammonia by Thiamet-G was not dependent on decreased liver damage. Together, these results

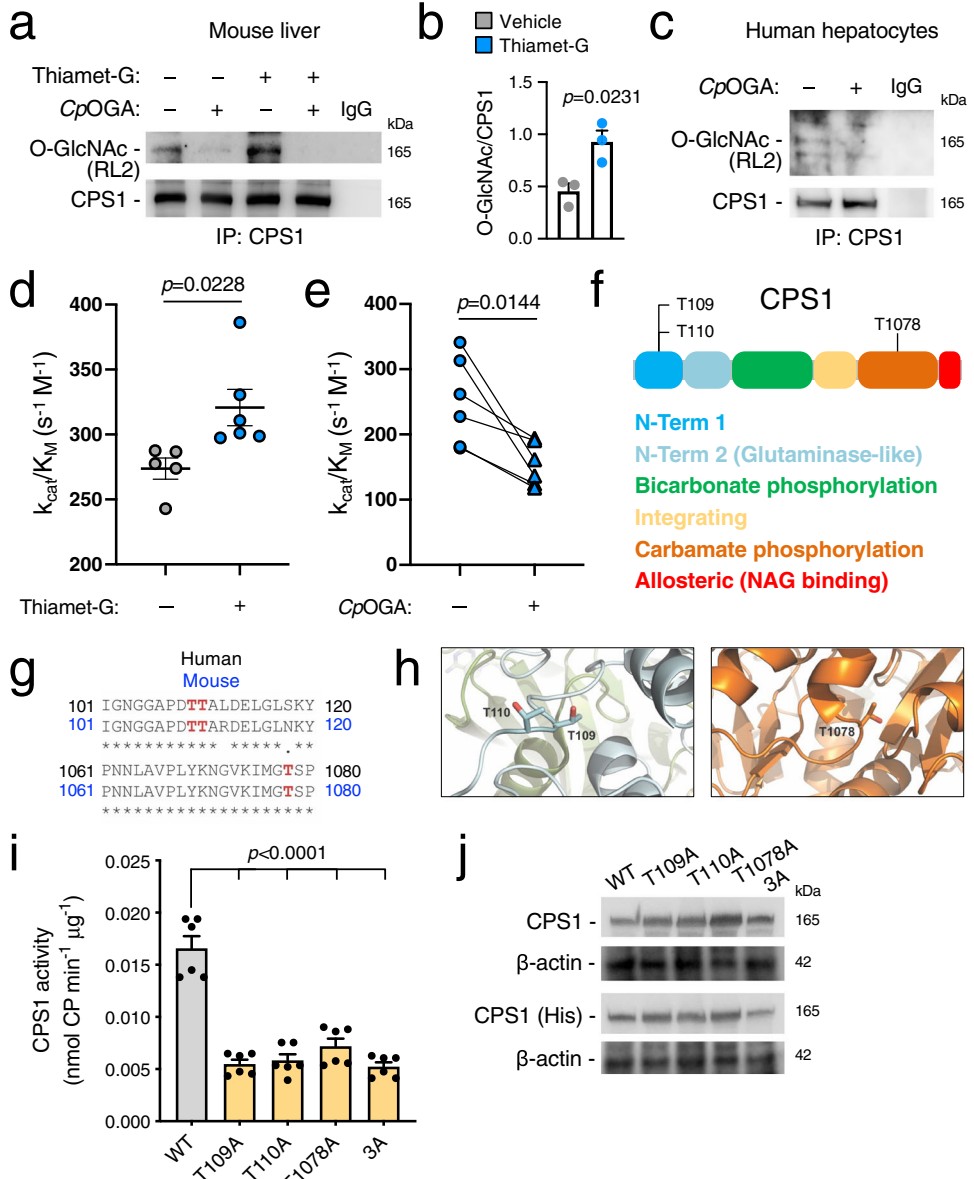

**Fig. 4 | CPS1 O-GlcNAcylation improves enzyme kinetics for ammonia.**
**a** Immunoprecipitated CPS1 is O-GlcNAcylated in livers of C57BL/6 wild-type (WT) mice i.p. injected with vehicle (PBS) or Thiamet-G (40 mg/kg for 2 days). Pre-incubation of liver lysates with the bacterial OGA orthologue from *Clostridium perfringens* (*Cp*OGA) that specifically removes the O-GlcNAc resulted in loss of CPS1 immunoreactivity detected by the anti-O-GlcNAc antibody (RL2). **b** Densitometric quantifications of bands detected with the O-GlcNAc antibody on immunoprecipitated CPS1. Data are representative of three independent experiments. $p = 0.0231$ (Unpaired *t*-test). **c** Immunoprecipitated CPS1 is O-GlcNAcylated in human hepatocytes. **d** CPS1 specificity constant (kinetic efficiency) for ammonia measured in livers harvested from WT mice after treatment with Thiamet-G (40 mg/kg, i.p. for 2 days) ($n = 6$ mice/group) or vehicle ($n = 5$ mice/group). $p = 0.0228$ (Unpaired *t*-test). **e** Analysis of CPS1 kinetic efficiency for ammonia in livers of WT mice treated

with Thiamet-G (40 mg/kg, i.p. for 2 days) after incubation of liver lysates with either *Cp*OGA or the corresponding buffer ($n = 6$). $p = 0.0144$ (Paired *t*-test). **f** Schematic diagram of CPS1 protein highlighting the different enzyme domains and the O-GlcNAc sites identified in this study (not drawn to scale). **g** Sequence alignment shows the highly conserved O-GlcNAcylated Thr109, Thr110 and Thr1078 (in red) between *Mus musculus* (Mouse) and *Homo sapiens* (Human). **h** 3D structures of CPS1 showing the localization of the identified Thr sites that are O-GlcNAcylated; colors match the different domains depicted in panel **f** [PDB ID code: 5DOU]. **i** Catalytic activity of human mature CPS1, either WT or mutant, expressed in a baculovirus/insect cell system ($n = 6$/group). $p < 0.0001$ (One-way ANOVA). **j** Western blot for WT and mutant CPS1 in insect cell lysates. β-actin was used as loading control. All values are shown as averages ± SEM. IP immunoprecipitation, Pi post-injection, 3A triple mutant. Experiments in panel **j** were performed twice.

suggest that pharmacological inhibition of OGA might be effective for therapy of hyperammonemia occurring in both inherited and acquired liver disorders.

## Discussion
The HBP is an important pathway for sensing metabolic status that involves several molecules, such as glucose, acetate, uridine, and glutamine. Glutamine and uridine are usually increased during

hyperammonemia, and they are also required for the synthesis of UDP-GlcNAc, a high-energy sugar donor that is transferred onto Ser or Thr residues of intracellular target proteins by OGT. O-GlcNAcylation is a fast-cycling PTM that plays important roles in liver metabolism[11]. In the present study, we describe a key role of O-GlcNAcylation in waste nitrogen removal by regulating the function of CPS1, the rate-limiting enzyme in ureagenesis (Fig. 5g). In addition, we provided clinically relevant data supporting the efficacy

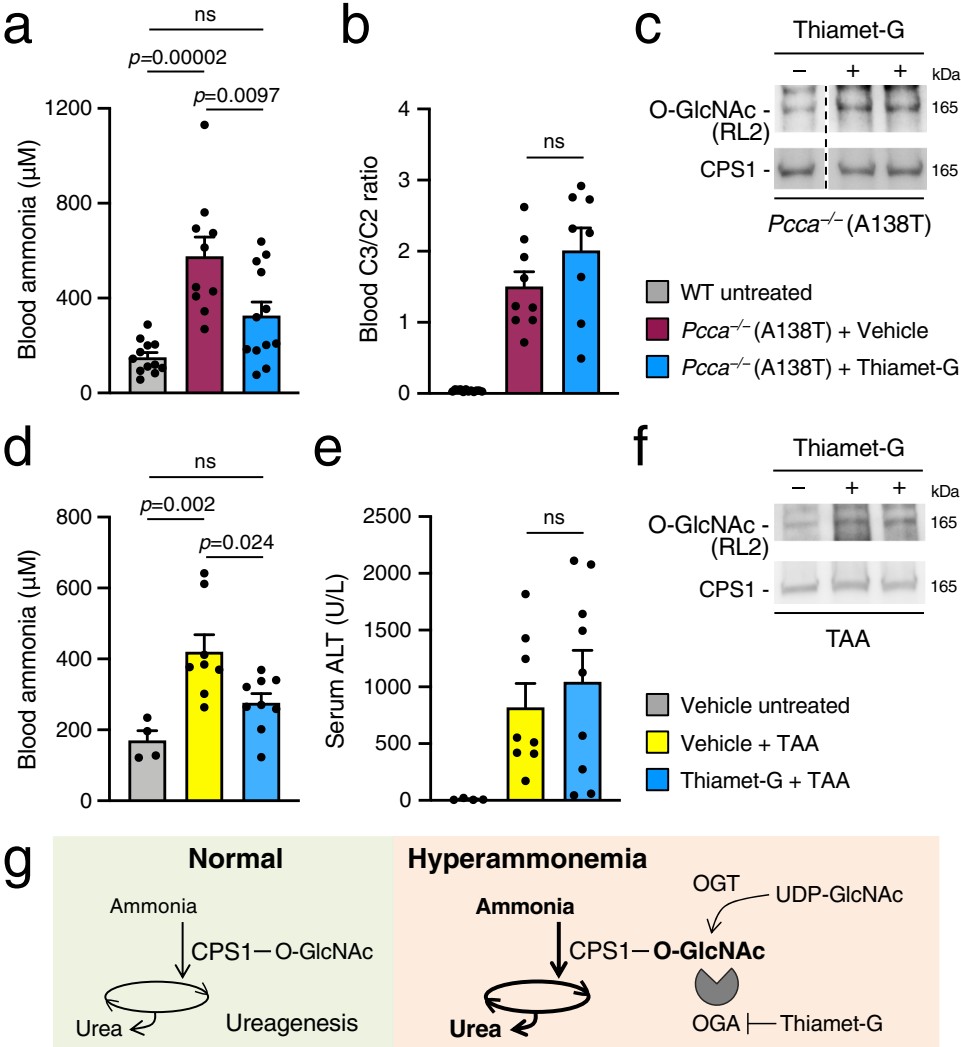

**Fig. 5 | OGA inhibition reduced hyperammonemia in mouse models of both inherited and acquired liver diseases. a** Blood ammonia in *Pcca*−/− (A138T) mice injected with Thiamet-G (40 mg/kg for 5 days) (*n* = 12) or vehicle (*n* = 10). Age-matched C57BL/6 wild-type (WT) untreated mice were used as controls (*n* = 12). *p* = 0.00002, *p* = 0.0097 (one-way ANOVA). **b** Ratio of propionylcarnitine (C3) to acetylcarnitine (C2) in sera of *Pcca*−/− (A138T) mice treated with Thiamet-G (*n* = 8) or vehicle (*n* = 9). Age-matched WT untreated mice were used as controls (*n* = 12) (One-way ANOVA). **c** Increased O-GlcNAcylation of CPS1 in livers of *Pcca*−/− (A138T) mice i.p. injected with Thiamet-G. Samples were run on the same gel but were non-contiguous. **d** Blood ammonia in WT mice treated with Thiamet-G (40 mg/kg, i.p. for 5 days) and/or thioacetamide (TAA; 250 mg/kg, i.p. 24 h before sacrifice) or with vehicle. Thiamet-G + TAA *n* = 9; Vehicle + TAA *n* = 8, Vehicle untreated *n* = 4. *p* = 0.002, *p* = 0.024 (One-way ANOVA). **e** Serum alanine transaminase (ALT) in WT mice treated with Thiamet-G and/or TAA or with vehicle. Thiamet-G + TAA *n* = 9; Vehicle + TAA *n* = 8, Vehicle untreated *n* = 4. **f** Increased O-GlcNAcylation of CPS1 in livers of TAA-treated mice i.p. injected with Thiamet-G. **g** Model for CPS1 O-GlcNAcylation in ureagenesis. Under normal conditions: ammonia is cleared by the synthesis of urea through the urea cycle and CPS1 is the rate-limiting step. During hyperammonemia, increased availability of UDP-GlcNAc is associated with greater O-GlcNAcylation of CPS1, enhanced enzyme catalytic efficiency for ammonia and overall ureagenesis, thus resulting in increased ammonia detoxification. Enhancement of CPS1 O-GlcNAcylation by Thiamet-G-mediated inhibition of OGA can be exploited to reduce hyperammonemia. All values are shown as averages ± SEM. ns, no statistically significant difference. Experiments in panels **c** and **f** were performed twice.

of OGA inhibitors for therapy of hyperammonemia due to both genetic and acquired disorders.

PTM of proteins regulate several metabolic processes by affecting protein abundance, stoichiometry, interaction, transport, and localization. Mechanistically, PTM provides rapid and largely reversible means of modulating enzymatic activity in response to various conditions. These modifications can either increase or decrease enzyme activity to face the changing needs of cells and tissues. To meet the ammonia detoxification demand, ureagenesis has to be highly regulated. However, few mechanisms regulating ureagenesis have been reported so far. Among these, protein acetylations on CPS1[23,24], ornithine transcarbamylase (OTC)[28], and argininosuccinate synthetase 1 (ASS1)[29] all have inhibitory effects on ureagenesis. Moreover, CPS1 is

hyperacetylated in obese mice[30], but it has also been found to be modified by lysine malonylation, succinylation, and glutarylation[25,31]. In addition, CPS1 can be modified by fatty acylation on cysteine residues[32] and nitration on tyrosine residues[33], both resulting in reduced enzyme activity. Here, we found that in contrast to all previously identified PTM, O-GlcNAcylation of CPS1 positively regulates the enzymatic function, resulting in increased ureagenesis. Further studies are required to establish the physiological role of CPS1 O-GlcNAcylation, as well as the interplay with other PTM regulating CPS1 function.

CPS1 O-GlcNAcylation was previously reported in livers of rodents[34,35]. In rats, CPS1 was found to be O-GlcNAc-modified at Ser537, Ser1331, and Thr1332[34] whereas in BALB/c mice at Ser872[35]. Compared to these previous studies, we identified three previously unreported

sites of CPS1 O-GlcNAcylation (Thr109, Thr110, and Thr1078), whereas none of the previously identified sites were detected[34,35]. At least one of these sites (Thr109) appears to be also phosphorylated, suggesting a crosstalk between O-GlcNAcylation and phosphorylation in regulation of CPS1 activity. However, further studies are needed to investigate the reciprocal regulation of CPS1 activity by these two PTMs. Moreover, a recent study found that O-GlcNAcylation at the sites previously described has an inhibitory effect on CPS1 enzymatic activity[36]. The reasons for this discrepancy are unclear. However, differences in species, strains, environmental and experimental conditions, and/or sample preparation might be involved.

CPS1 is a 1500 amino acid enzyme organized into multiple domains that catalyzes the entry of ammonia into the urea cycle through a three-step irreversible reaction. For the ~40-kDa N-terminal region of CPS1, no functions are known, and this domain of CPS1 has been subdivided into the N-terminal-1 and N-terminal-2 subdomains, the latter corresponding to an ancestral but inactive glutaminase-like domain[26]. Despite being located outside the catalytic domain, removal of the full N-terminal domain drastically reduced CPS1 enzymatic activity[37], suggesting that this domain plays a role at structural level, possibly affecting proper conformation of the catalytic centers. Moreover, pathogenic variants affecting this enzyme domain have been found in patients with CPS1 deficiency presenting both in the neonatal period or later in life[38]. Elegant studies have shown that pathogenic variants mapping to the N-terminal subdomain result in a mild reduction of enzyme activity[39,40], suggesting that the N-terminal domain is important in the regulation of enzymatic catalytic power. Moreover, the crystal structure of the human CPS1 protein revealed cavities and tunnels for ammonia and carbamate transport across the enzyme[26] and a tunnel was predicted in the N-terminal domain, at the molecular surface between Glu82 and His142 residues, that is likely involved in the intake of external ammonia[26]. Interestingly, we found that in this N-terminal domain of CPS1 where the tunnel for ammonia incorporation is predicted to start, Thr109 and Thr110 are O-GlcNA-cylated, suggesting that this PTM could regulate ammonia uptake by CPS1. Consistently, we found that O-GlcNAcylation enhances CPS1 kinetic efficiency for ammonia. The other O-GlcNAcylated residue Thr1078 is located within the carbamate phosphorylation domain, and O-GlcNAcylation of this residue could play a role in remodeling and stabilization of the carbamate transport tunnel[26]. Although no pathogenic variants affecting Thr109, Thr110 or Thr1078 have been reported in patients with CPS1 deficiency[38,41], missense variants involving all three Thr residues are very rare with only one heterozygote over more than 25,000 control alleles for Thr109 and Thr1078 and none for Thr110. In addition, no homozygotes are found in gnomAD for these variants and site-directed mutagenesis of these Thr residues resulted in reduction of CPS1 enzyme activity. Altogether, this suggests that these three Thr residues are highly intolerant to variations and thus, are predicted to play an important role in protein function.

Recent findings showed cross-talks between ureagenesis and bile acid metabolism or autophagy[14,42], that are both regulated in the liver by O-GlcNAcylation[43,44]. Therefore, it is possible that OGA inhibition by Thiamet-G may have additional effect on ammonia detoxification by modulating these pathways.

The findings of this study have therapeutic potential for treatment of CPS1 deficiency and other UCD, such as N-acetyl-glutamate synthase (NAGS) deficiency. Although we do not expect Thiamet-G, or OGA inhibitors in general, to be effective for therapy of patients carrying bi-allelic loss-of-function pathogenic variants resulting in complete deficiency of CPS1 enzyme activity, they could be effective in patients harboring missense variants resulting in increased protein degradation and/or reduced catalytic activity[41]. To be fully activated, CPS1 requires its allosteric activator NAG, which is generated by NAGS[45]. Subjects with late-onset CPS1 deficiency can be treated with N-carbamyl-L-glutamate, a deacylase-resistant analogue of NAG

developed to treat NAGS deficiency[46]. However, treatment with N-carbamyl-L-glutamate reduced residual activity of the enzyme carrying some pathogenic variants[47], and Thiamet-G or other OGA inhibitors might be alternative therapeutic agents in these cases or an adjunctive treatment to achieve rapid and more effective control of blood ammonia in patients carrying CPS1 hypomorphic variants. Nevertheless, enhancement of CPS1 O-GlcNAcylation is unlikely to be effective in other UCD due to deficiencies in enzymes that are downstream of CPS1 and they could even be worsened by increased incorporation of ammonia into toxic urea cycle intermediates.

Systemic blood ammonia increases during acute decompensation associated with inherited conditions with secondary deficiency of CPS1 activity such as organic acidurias[48], whereas hyperammonemia occurring in acquired liver failure is not due to deficiencies of specific urea cycle enzymes but it is rather due to a global reduction of ureagenesis secondary to hepatocyte loss. Results of this work suggest that both these conditions might be treated by OGA inhibitors. Similarly, non-alcoholic steatohepatitis associated with reduced gene and protein expression of CPS1 resulting in hyperammonemia[49] might also be a target of OGA-inhibitors. Although OGA inhibitors are not yet approved as therapies in humans, preclinical studies have shown efficacy of Thiamet-G-mediated enhancement of protein O-GlcNAcylation in Tau-induced neurodegeneration, and MK-8719, an optimized analog of Thiamet-G, was well tolerated in Phase I clinical trials[50,51]. Nevertheless, O-GlcNAcylation regulates several cellular processes and disruption of O-GlcNAc homeostasis has been also implicated in the pathogenesis of human diseases[52], suggesting that sustained OGA inhibition might be toxic. Interestingly, it has been proposed that cells can tolerate mild increases or reductions in global O-GlcNAcylation levels, as long as they are within an optimal zone, that is maintained by mutual autoregulatory loops between OGT and OGA expression and activity[11]. Therefore, although long-term treatments might be associated with toxicity or reduced efficacy because of such regulatory loops, short-term OGA inhibition might still be effective for treatment of life-threatening acute hyperammonemia due to either inherited or acquired diseases.

Cornerstones of treatment for UCD, including CPS1 deficiency and other inherited disorders with hyperammonemia, are restriction of dietary protein intake, scavenging ammonia drugs, and orthotopic liver transplantation. However, available therapeutic options are often unsatisfactory and thus, better treatments have been actively investigated[5]. Drugs acting on PTM have potential for treatment of these disorders as they can increase the enzyme residual activity. In contrast to disorders due to complete deficiency of enzyme activity, diseases due to partial enzyme defects are expected to be more susceptible to treatments with small molecule drugs that increase residual enzyme activity. The present proof-of principle study highlighted the efficacy of drugs increasing protein O-GlcNAcylation as candidates to induce rapid and clinically relevant reductions of hyperammonemia.

Through association studies, *CPS1* polymorphisms have been implicated in coronary artery disease and persistent pulmonary hypertension of the newborn[18], possibly by reducing arginine-derived nitric oxide metabolites. Moreover, they have been involved in weight loss maintenance and neuropsychiatric disorders[18]. Therefore, modulation of CPS1 activity by OGA inhibition might also have therapeutic implications in a wide range of multifactorial disorders.

In cancer, CPS1 overexpression supports pyrimidine synthesis to promote sustained cell growth or to prevent the build-up of toxic levels of intra-tumoral ammonia[53,54]. Interestingly, HBP was upregulated in aggressive human non-small-cell lung cancers subtypes harboring *CPS1* gain-of-function variants[55]. Because CPS1 overexpression is a driver of drug-resistant cancers associated with poor outcomes[56], targeted CPS1 inhibitors have been proposed as anti-

cancer therapy[57]. Based on the findings of this study, OGT inhibitors might be an alternative or adjunctive treatment for cancers with CPS1 overexpression.

In conclusion, our data show that hepatic O-GlcNAcylation enhances CPS1 catalytic efficiency for ammonia and promotes urea-genesis, thus indicating OGA as a novel therapeutic agent for therapy of both genetic and acquired diseases resulting in hyperammonemia.

# Methods

## Mouse procedures

All mouse procedures were performed in accordance with regulations and were authorized by the Italian Ministry of Health; the National Institutes of Health and the Institutional Animal Care and Use Committee of David Geffen School of Medicine at UCLA (Los Angeles, United States); and the Spanish Law of Animal Protection. Investigators were not blinded to allocation during experiments and outcome assessment. Male wild-type (WT) 6-week-old C57BL/6 mice (Charles River Laboratories, Calco, Italy) were randomly assigned to treatment groups. Mice were housed in ventilated cages, maintaining a temperature of 22 °C (±2 °C), relative humidity of 50% (±5%), and 12-h light/12-h dark cycle, and receiving a standard chow diet and water ad libitum. For induction of acute hyperammonemia, mice were starved overnight before the intraperitoneal (i.p.) injection of 10 mmol/kg of $^{15}$N-labeled ammonium chloride (98% enriched in $^{15}$N, Sigma, Cat# 299251) dissolved in water. For liver-specific gene transfer of OGT, WT mice were intravenously (i.v.) injected in the retro-orbital plexus with a dose of $1 \times 10^{13}$ genome copies (GC)/kg of the AAV-OGT or AAV-GFP, and ureagenesis was evaluated four weeks post-vector administration by stable isotope tracing after i.p. injections of $^{15}$N-labeled ammonium chloride (2 mmol/kg) or sodium acetate-1-$^{13}$C (99% enriched in $^{13}$C, Sigma, Cat# 279293) (55 mg/kg). For liver-specific overexpression of OGA, WT mice were injected i.v. with a dose of $1 \times 10^{13}$ GC/kg of the AAV-OGA or AAV-GFP. The i.p. injections of ammonia in these mice was performed 4 weeks post-vector injections. All vectors were prepared in sterile pharmaceutical-grade saline. Hepatic *knock-down* of OGA in WT mice was induced by the Invivofectamine 3.0 system (ThermoFisher, Cat# IVF3001). The siRNA duplex solution was prepared according to the manufacturer's instructions, and injected i.v. at a final concentration of 1 mg/kg. Mice received a mix of Ambion pre-designed siRNA against murine OGA (*MGEA5* gene: ID: #167691 and #167693, Invitrogen, Cat# AM16708). Sense (5′–3′): GGGAGAAUAUCAUGACCAAtt; Antisense (5′–3′): UUGGUCAUGAUAUUCUCCCtg, and Sense (5′–3′): GCAGUUGUUUUGACUUACUtt; Antisens2e (5′–3′): AGUAAGU-CAAAACAACUGCtt. Control mice were injected with the Silencer™ Select Negative Control No. 1 siRNA (Invitrogen, Cat# 4390843). The i.p. injections of ammonia were performed 6 days after siRNA administrations. All i.v. injections were performed by retro-orbital injections under isoflurane anesthesia. ST045849 (Tocris, Cat# 6775; 40 mg/kg) was re-suspended in 10% Dimethyl Sulfoxide (DMSO), 5% Tween-20 in saline solution and delivered by i.p. injection 18 h and 2 h prior to the ammonia i.p. injections. Thiamet-G (Sigma-Aldrich, Cat# SML0244; 40 mg/kg) was dissolved in phosphate-buffered saline (PBS) solution and delivered by i.p. injections for two days prior to the ammonia i.p. injections. PUGNAc (Sigma-Aldrich, Cat# A7229; 7 mg/kg) was dissolved in PBS solution and delivered by i.p. injections for two days prior to the ammonia i.p. injections. Thioacetamide (Sigma-Aldrich, Cat# 172502) was re-suspended in saline solution and injected i.p. at the dose of 250 mg/kg. Four days prior to the administration of TAA, mice were injected i.p. daily with Thiamet-G at a dose of 40 mg/kg. Hypomorphic *Pcca*$^{−/−}$ (A138T) mice[27] maintained on an C57BL/6 background were genotyped as previously reported[58]. Four-week-old male and female mice were included. Mice were injected i.p. daily with 40 mg/kg/day of Thiamet-G for four days prior to sacrifice (24 h after the last injection). Control mice were injected with vehicle only. Blood samples were collected by retro-

orbital bleedings and livers were harvested in mice sacrificed by cervical dislocation.

## Adeno-associated viral (AAV) vectors

AAV2/8 vectors encoding mouse OGT, mouse OGA, or GFP under the control of the HLP promoter[59] were generated. The murine-tagged OGT (OriGene, Cat# MR211521) and murine-tagged OGA (OriGene, Cat# MR211167) encoding sequences were cloned into the pHLP-GFP plasmid for the generation of the AAV vectors. All vectors were produced by the AAV Vector Core of Telethon Institute of Genetics and Medicine (TIGEM), as previously described[60]. Physical titers of the viral vector preparations (genome copies/ml) were determined by real-time PCR quantification using TaqMan (Applied Biosystems, Foster City, CA) and dot-blot analysis[60]. The final titer of each prep was calculated as the average between the PCR quantification and dot-blot results.

## Biochemical measurements

Blood ammonia concentrations were measured by an ammonia colorimetric assay kit (BioVision Incorporated, Cat# K370-100) according to the manufacturer's instructions. Blood and liver glutamine content were measured by colorimetric assay kit (BioVision Incorporated, Cat# K556-100). Serum ALT was analyzed with a scil VitroVet (scil animal care company S.r.l., Treviglio, Italy). Liver uridine was determined by high-resolution proton nuclear magnetic resonance ($^{1}$H-NMR) spectroscopy, using a Bruker AVANCE III HD-600 spectrometer (Bruker BioSpin GmbH, Rheinstetten, Germany). Liver content of UDP-GlcNAc was determined by hydrophilic interaction liquid chromatography (HILIC) at the Academic Medical Center (Amsterdam, the Netherlands)[61]. Serum acylcarnitines were analyzed by tandem mass spectrometry at the Centro de Diagnostico de Enfermedades Moleculares (CEDEM), Universidad Autonoma, Madrid as previously reported[62]. Measurements of blood levels of $^{15}$N-labeled urea was performed by either $^{15}$N-NMR spectroscopy or gas chromatography–mass spectrometry (GC-MS) as previously described[14,63], using a Bruker AVANCE TM III HD-400 spectrometer (Bruker BioSpin GmbH, Rheinstetten, Germany), or a mass spectrometer Polaris Q Trace GC Ultra (Thermo Scientific, Switzerland), respectively. The $^{13}$C-labeled urea in sera was quantified by isotope ratio-mass spectrometer (IRMS) analysis using a Finnigan Delta Plus isotope ratio-mass spectrometer (Thermo Fisher Scientific, Waltham, MA) at the Metabolic Core of The Children's Hospital of Philadelphia (Philadelphia, PA).

## Immunohistochemistry

Mouse liver specimens were fixed in 4% paraformaldehyde for 12 h, stored in 70% ethanol, and embedded into paraffin blocks. For immunohistochemistry, 5-µm thick sections were rehydrated and permeabilized in PBS with 0.5% Triton (Sigma-Aldrich) for 20 min. Antigen unmasking was performed in 0.01 M citrate buffer in a microwave oven. Next, sections underwent blocking of endogenous peroxidase activity in methanol/1.5% $H_2O_2$ (Sigma-Aldrich) for 30 min and were incubated with blocking solution (3% BSA [Sigma-Aldrich], 5% donkey serum [Millipore], 1.5% horse serum [Vector Laboratories] 20 mM MgCl$_2$, 0.3% Triton [Sigma-Aldrich] in PBS) for 1 h. Sections were incubated with primary antibody anti-O-GlcNAc (RL2) (Abcam, Cat# ab2739, dilution 1/200) overnight at 4 °C and with universal biotinylated horse anti-mouse IgG secondary antibody (dilution 1/200) (Vector Laboratories, PK-620) for 1 h. Biotin/avidin-HRP signal amplification was achieved using ABC Elite Kit (Vector Laboratories, PK-6200) according to the manufacturer's instructions. 3,3′-diaminobenzidine (Vector Laboratories) was used as peroxidase substrate. Mayer's hematoxylin (Bio-Optica) was used as counter-staining. Sections were de-hydrated and mounted in VECTASHIELD® (Vector Laboratories). Paraffin-embedded liver sections were stained with hematoxylin and eosin (H&E). Sections were mounted in Mowiol

(Sigma-Aldrich) mounting medium. Image capture was performed using an Axio Scan Z.1 microscope (Zeiss).

## Subcellular fractionation

Liver samples were homogenized using a Teflon pestle and mortar and suspended in mitochondrial isolation buffer (250 mM sucrose, 20 mM HEPES, 10 mM KCl, 1.5 mM MgCl₂, 1 mM EDTA, 1 mM EGTA, pH 7.4) supplemented with protease and phosphatase inhibitor cocktails (Complete and PhosSTOP Roche, Roche Diagnostics, Basel, Switzerland). Lysates were centrifuged at $1000 \times g$ for 10 min at 4 °C to pellet the nuclei while mitochondrial and cytosolic fractions were contained within the supernatant. Pellets containing nuclei were washed twice with PBS and centrifuged at $800 \times g$ for 10 min. Pellets containing nuclei were re-suspended in nuclear lysis buffer (1.5 mM MgCl₂, 0.2 mM EDTA, 20 mM HEPES, 0.5 M NaCl, 20% glycerol, 1% Triton X-100) and incubated on ice for 30 min and then sonicated at 3 Å for ~10 s (for 3 times). Samples were then centrifuged for 15 min at $16,000 \times g$. The supernatant contained the enriched nuclear fraction. Supernatants containing mitochondrial and cytosolic fractions were re-centrifuged twice at $16,000 \times g$ for 20 min at 4 °C to pellet the mitochondria. The resulting supernatant was the enriched cytosolic fraction. The mitochondrial pellets were then resuspended in a fourfold dilution of mitochondrial isolation buffer and centrifuged at $16,000 \times g$ for 20 min at 4 °C twice. The mitochondrial pellets were resuspended in a one-fold dilution of mitochondrial isolation buffer and then sonicated to yield the mitochondrial fraction.

## GFPT1 enzyme activity assay

GFPT1 enzyme activity was measured using the glutamate dehydrogenase (GDH) method in total liver protein extracts, as reported previously[64] with minor modifications. Briefly, 20 µg of total liver protein extract (in lysis GFPT1 buffer: 50 mM Tris, 5 mM EDTA, 5 mM GSH, 5 mM glucose-6-phosphate, 50 mM KCl, pH 8.0) were incubated at 37 °C for 90 min with the GDH reaction buffer (50 mM Tris, 6 mM glutamine, 0.8 mM fructose-6-phosphate, 0.3 mM APAD, 50 mM KCl, 0.1 mM KH₂PO₄, 6 Units of GDH, pH 8.0). Glutamate production was determined by measuring the absorbance at 370 nm. Absorbance of the blank reaction mixture were used as reference. All reagents were purchased from Sigma-Aldrich.

## Western blotting

Liver specimens were homogenized in RIPA buffer in the presence of a complete protease inhibitor cocktail (Sigma), incubated for 20 min at 4 °C and centrifuged at $16,800 \times g$ for 10 min. Pellets were discarded and cell lysates were used for Western blots. Total protein concentration in cellular extracts was measured using the Bradford Reagent (Bio-Rad, Hercules, California, USA). Protein extracts were separated by SDS–PAGE and transferred onto polyvinylidene difluoride (PVDF) membranes. Blots were blocked with TBS-Tween-20 containing 5% non-fat milk for 1.0 h at room temperature followed by incubation with primary antibody overnight at 4 °C. The primary antibodies used are summarized in Supplementary Table 1. Anti-O-GlcNAc antibodies validation is shown in Supplementary Fig. 11. Proteins of interest were detected with horseradish peroxidase (HRP)-conjugated goat anti-mouse or donkey anti- rabbit IgG antibody (GE Healthcare, dilution 1/5000). Peroxidase substrate was provided by ECL Western Blotting Substrate kit (Pierce). Densitometric analyses of the Western blotting bands were performed using ImageJ Software (Fiji 2). Uncropped blot scans are shown in Supplementary Fig. 12.

## Identification of O-GlcNAcylated proteins in mouse liver

Analysis of mouse liver protein O-GlcNAcylation was performed by immunoprecipitation and mass spectrometry (MS) analysis. Liver lysates were prepared in cold RIPA lysis buffer (50 mM Tris-HCl, 150 mM NaCl, 1% Triton X-100, 1 mM EDTA, 0.1% SDS, pH 7.4) in the

presence of protease inhibitors (Sigma-Aldrich) and Thiamet-G (1 mM), incubated for 20 min on ice, and centrifuged at $16,800 \times g$ for 10 min. Supernatants were collected and protein concentration was determined by the Bradford Reagent (Bio-Rad). Next, lysates (750 µg) were incubated with 7.5 µg anti-O-GlcNAc (RL2) (Abcam, Cat# ab2739) antibody overnight at 4 °C under rotation. As negative controls, lysates were incubated with anti-mouse IgG isotope (Thermo Fisher Scientific, Cat# 31903, ratio 1 µg IgG:100 µg lysate). The following day, lysates were incubated with 60 µL of magnetic beads Dynabeads™ Protein G (Thermo Fisher Scientific, Cat# 1004 D) for 2 h at 4 °C under gentle rotation. Using a DynaMag™ magnet (Thermo Fisher Scientific, Cat# 12321D), the beads were then washed three times with 1 mL of PBS containing 0.02% of Tween-20 at 4 °C for 5 min under rotation, and immunoprecipitated CPS1 was eluted in Laemmli buffer (62.5 mM Tris-HCl, 2% SDS, 10% glycerol, 20 mM DTT, pH 6.8) and separated by SDS-PAGE. Gel was stained with Coomassie blue dye and processed for MS analysis. Gel lanes were processed according to STAGE-diging protocol[65]. Briefly, the entire lanes were transferred into the STAGE-diging p1000 tip filled with a double C18 Empore Disk (3 M, Minneapolis, MN) plug and submitted to reduction with 10 mM dithiothreitol (DTT), alkylation with 55 mM iodoacetamide (IAA) and Trypsin digestion overnight. Digestion solutions were acidified and desalted passing through the C18 plugs then eluted peptides were dried in a Speed-Vac and resuspended in solvent A (2% acetonitrile, 0.1% formic acid). For liquid chromatography–tandem MS (nLC–MS/MS) analysis, 2 µL for each digested sample were injected in technical replicates on a nanoLC–MS/MS quadrupole Orbitrap Q Exactive HF mass spectrometer. Peptide separation was achieved on a linear gradient of 32 min, starting from 95% of solvent A and ramping to 50% Solvent B (80% acetonitrile, 0.1% formic acid) in 23 min and from 50 to 100% Solvent B in 2 min at a constant flow rate of 0.25 µl min⁻¹ on a UHPLC Easy-nLC 1000 (Thermo Scientific). The LC system was connected to a 25 cm fused-silica emitter of 75 µm inner diameter (New Objective), packed in house using a high-pressure bomb loader (Proxeon) with ReproSil-Pur C18-AQ 1.9 µm beads (Dr. Maisch). MS data were acquired in data-dependent mode, with a top15 method for HCD fragmentation. Survey full scan MS spectra (300–1750 Th) were acquired in the Orbitrap with 60,000 resolution, AGC target 1e6, IT 120 ms. For HCD spectra the resolution was set to 15,000, AGC target 1e5, IT 120 ms; normalized collision energy 28, isolation width 3.0 m/z with a dynamic exclusion of 5 s. Proteins were identified processing Raw files with MaxQuant version 1.5.2.8 searching against the database uniprot_cp_mouse in which Trypsin enzyme specificity and up to two missed cleavages were allowed; carbamidomethylation of cysteine was set as fixed modification while oxidation of methionine and acetylation of N-term protein were set as variable modifications. Mass deviation for MS/MS peaks was 20 ppm. The peptides and protein false discovery rates (FDR) were set to 0.01; the minimal length required for a peptide identification was six amino acids; a minimum of two peptides and at least one unique peptide was required for high-confidence protein identification. The identified proteins were filtered to eliminate known contaminants.

## CPS1 immunoprecipitation

Mouse liver lysates (500 µg), insect cells lysates (500 µg), or commercially available human hepatocytes (Cryo Human Hepatocytes from GIBCO, Source: Single Donor, Lot# HU8300) (200 µg) were incubated with anti-CPS1 antibody (Abcam, Cat# ab45956) (ratio 1 µg antibody:100 µg lysate) overnight at 4 °C under rotation. For experiments with cryopreserved human hepatocytes, cells (4–8 million) were stored at −135 °C, thawed the day of sample preparation, and lysed in cold RIPA lysis buffer in the presence of protease inhibitors and Thiamet-G (1 mM), incubated for 20 min on ice, and centrifuged at $16,800 \times g$ for 10 min. As negative control, lysates were incubated with anti-rabbit IgG isotope (Thermo Fisher Scientific, Cat# 31235). The

following day, lysates were incubated with 40 μL of magnetic beads Dynabeads™ Protein G (Thermo Fisher Scientific) for 2 h at 4 °C under gentle rotation. Using a DynaMag™ magnet (Thermo Fisher Scientific), the beads were then washed 3 times with 1 mL of PBS containing 0.02% of Tween-20 at 4 °C for 5 min under rotation, and immunoprecipitated CPS1 was eluted in Laemmli buffer (62.5 mM Tris-HCl, 2% SDS, 10% glycerol, 20 mM DTT, pH 6.8), separated by SDS-PAGE and analyzed by Western blot with antibodies of interest. To induce O-GlcNAc hydrolysis, protein lysates were pre-incubated for 1.0 h at 30 °C with recombinant OGA from *C. perfringens*[19].

### Production of mutant human and mouse CPS1

Thr were mutagenized into Ala in the pFastBac-CPS1[40] plasmids using the Quikchange II XL Site-directed Mutagenesis Kit (Agilent Technologies, Cat#200521). The pCMV-*Cps1*-Myc plasmid was generated by cloning the murine *Cps1* coding sequence (Blue Heron Biotech, LLC) into the pCMV-Myc (Thermo Fisher Scientific). Primers used for mutagenesis are listed in Supplementary Table 2. The correctness of the constructs, the presence of the desired mutations, and the absence of unwanted mutations were confirmed by sequencing (Eurofins Genomics). Human mature CPS1, either wild-type or mutant forms, were expressed in a baculovirus/insect cell system and enzyme activity was assessed in crude extracts[40].

### Ammonia removal in cells transfected with plasmids expressing CPS1

Human Huh-7 hepatic cells were cultured in Dulbecco's modified Eagle's medium supplemented with 10% fetal bovine serum and 1% penicillin/streptomycin at 37 °C in 5% $CO_2$. Routine testing for Mycoplasma was not performed. Cells were transfected with either the wild-type (pCMV-Cps1$^{WT}$-Myc) or mutagenized (pCMV-Cps1$^{3A}$-Myc) plasmids using LipoD293™ DNA transfection reagent (SignaGen Laboratories; Cat# SL100668) according to manufacturer's instructions. Forty-eight hours after transfection, cells were incubated with ammonium chloride (1 mM) and L-ornithine (1 mM) for 24 h. Culture medium was then collected and centrifuged at 1000 × g for 5 min to obtain a cell-free supernatant for measurement of ammonia concentrations by an ammonia colorimetric assay kit (BioVision Incorporated, Cat# K370-100), and cells were harvested and processed for Western blot analysis.

### CPS1 enzyme assays

CPS1 enzymatic activity on mouse liver samples and recombinant proteins was determined by colorimetric assay, as previously described[40]. Briefly, the 10-min assay took place at 37 °C using an OTC coupled reaction, where carbamoyl phosphate was immediately converted to citrulline. For substrate or NAG allosteric activator kinetic analyses, the concentration of one assay component was varied while the rest were kept fixed at the saturating concentrations found in the standard reaction cocktail[40]. Thiamet-G (1 mM) was included in the lysis buffer when necessary. For probing O-GlcNAc removal and its effect on CPS1 kinetics, protein lysates were pre-incubated for 1.0 h at 30 °C with recombinant OGA from *C. perfringens*[19]. Control samples were incubated for 1.0 h at 30 °C with 25% glycerol in PBS. Data points were fitted to the Michaelis-Menten kinetic equation using GraphPad Prism software. CPS1 catalytic efficiency, computed for each substrate and NAG as the ratio of the catalytic constant $k_{cat}$ over the Michaelis constant $K_M$, was employed as a measure of the enzyme's overall ability to convert substrate to product, reflecting both binding and catalytic events.

### O-GlcNAc site mapping on CPS1

Livers were homogenized in immunoprecipitation lysis buffer (25 mM Tris, 150 mM NaCl, 1% NP-40, 5% Glycerol, 50 mM NaF, 5 mM Na Pyrophosphate, Protease inhibitors cocktail and 1 μM GlcNAcstatin G, pH 7.4) with 30 strokes in a glass homogenizer in ice. Lysates were kept on ice for 30 min and cellular debris were removed by centrifugation at

17,000 × g at 4 °C. Protein concentration was quantified using Pierce 660 nm assay, and 3 mg of lysate was incubated with 30 μg of antibody against CPS1 (Abcam, Cat# ab45956), or normal rabbit IgG as control (Cell Signaling Technology, Cat# 2729), overnight at 4 °C in an Eppendorf rotator. The following day, 50 μl of Dynabeads Protein G (Thermo Fisher) were washed two times in immunoprecipitation lysis buffer and added to the lysate-antibody complex for 1.0 h at 4 °C with rotation. Beads were then washed three times with immunoprecipitation lysis buffer by gently pipetting and resuspended in 100 μl of immunoprecipitation lysis buffer and transferred to a clean tube. Beads were then resuspended in 10 μl of premixed NuPAGE LDS Sample Buffer (Thermo Fisher, Cat# NP0007) and 20 μl of Elution Buffer (50 mM glycine pH 2.8) and heated for 10 min at 70 °C. Sample (including beads) was loaded on a 4–12% NuPAGE Bis-Tris gel (Invitrogen, Cat# NP0335PK2) and run until completion for MS. For MS, upon electrophoresis, gel was stained with Coomassie and thoroughly washed with MS grade water in a plastic tray to visualize CPS1 pulled protein. The band was then excised into cubes (ca. 1 × 1 mm) and transferred into a microcentrifuge tube. In-gel reduction, alkylation, and de-staining were performed as previously described[66]. Briefly, gel pieces were reduced with 10 mM DTT in 100 mM ammonium bicarbonate, alkylated in 55 mM of iodoacetamide in 100 mM ammonium bicarbonate and de-stained in 100 mM ammonium bicarbonate/acetonitrile (1:1 vol/vol) solution. In-gel digestion was performed overnight with 12 ng/μl Trypsin in 10 mM ammonium bicarbonate at 37 °C. Peptides were extracted with 100 ml of extraction buffer [1:2 (vol/vol) 5% formic acid/acetonitrile] to each tube and incubated for 15 min at 37 °C in a shaker. Peptides were then dissolved and resuspended in 0.1% (vol/vol) trifluoroacetic acid, and clean using ZipTip with 0.6 μL C18 resin (Millipore). Peptides were quantified with Pierce Quantitative Fluorometric Peptide Assay (Thermo Fisher, Cat. No 23290). Samples were analyzed using an Orbitrap Fusion™ Tribrid™ Mass Spectrometer (Thermo Scientific). Peptides were injected onto a 75 μm × 2 cm PepMap-C18 pre-column and resolved on a 75 μm × 50 cm RP- C18 EASY-Spray temperature-controlled integrated column-emitter (Thermo Scientific), using a four-hour multistep gradient from 5% B to 35% B with a constant flow of 200 nl min⁻¹. The mobile phases were: 2% acetonitrile incorporating 0.1% formic acid (solvent A) and 80% acetonitrile incorporating 0.1% formic acid (solvent B). The spray was initiated by applying 2.5 kV to the EASY-Spray emitter and the data were acquired under the control of Xcalibur software in a data-dependent mode using top speed and 4 s duration per cycle. The survey scan is acquired in the orbitrap covering the *m/z* range from 400 to 1400 Thomson with a mass resolution of 120,000 and an automatic gain control (AGC) target of $2.0 \times 10^5$ ions. HCD and EThcD MS/MS spectra were searched with Proteome Discoverer 2.4.1.15. Dynamic O-GlcNAc sites were searched as (HexNAc, +203.079 Da) at Ser and Thr residues.

### Statistical analyses

Data were analyzed using GraphPad Prism software (San Diego, CA, USA). A two-tailed paired Student's *t* test was performed when comparing the same group of mice at two different experimental conditions. Comparisons of continuous variables between two and more experimental groups were performed using the two-tailed unpaired Student's *t*-test or one-way ANOVA with Tukey's. Two-way ANOVA and Tukey's or Sidak's multiple tests were performed to compare two groups relative to two factors. No statistical methods were used to predetermine the sample size. Number of replicates is reported in the figure legends. Data are expressed as means ± SEM. *P* values < 0.05 were considered statistically significant.

### Reporting summary

Further information on research design is available in the Nature Research Reporting Summary linked to this article.

## Data availability

All data reported in the paper are included in the manuscript or available in the Supplementary information. Human CPS1 crystal structure obtained from the PDB database (https://www.rcsb.org/structure/5dou) was used. Proteomic data are available via ProteomeXchange[67] with identifier PXD024526. Source data are provided with this paper.

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

## Acknowledgements

We are grateful to Andrea Ballabio for critical reading of the manuscript and Phoebe Ashley-Norman (TIGEM Scientific Office) for manuscript editing. Measurement of UDP-GlcNAc was performed by the Amsterdam UMC Core Facility Metabolomics, specifically by Michel van Weeghel. Mass-Spectrometry Proteomics analysis was performed at the Cogentech Proteomics/MS Facility, Milano. AAV vector production was performed by the TIGEM AAV Vector Core. We thank Veronica Maffia (TIGEM) and Dominique Melck for technical assistance, and Pedro Ruiz-Sala from the diagnostic lab CEDEM for the carnitine determinations. This work was supported by grants of Fondazione Telethon Italy (to N.B.-P.), MIUR (PRIN2017 to N.B.-P.), of the Swiss National Science Foundation (grant 320030_176088 to J.H.), of the US National Institutes of Health grants (R21NS091654 and R01NS100979 both to G.S.L.), of the Spanish Ministry of Science and Innovation (PID2019-105344RB-I00/AEI/10.13039/501100011033 to L.R.D. and E.R.), and by a Wellcome Trust Investigator Award (110061 to D.M.f.v.A.).

## Author contributions

L.R.S. performed study concept and design, acquisition of data, analysis, interpretation of data, and wrote the manuscript; G.M. performed CPS1 kinetics studies; A.M.D.'A. and A.D.A. performed acquisition and analysis of data with the assistance of P.A.; I.B. performed site-directed mutagenesis; V.M.P. performed CPS1 O-GlcNAc mapping studies with the assistance of A.T.F.; V.R. helped with CPS1 enzyme assay; S.A. made cloning of AAV plasmids; E.N. performed injections into mice; D.P. and P.C. performed NMR studies supervised by A.M.; A.M.-P. performed studies in *Pcca^{-/-}* (A138T) mice supervised by E.R. and L.R.D.; M.N. performed studies in mice supervised by G.S.L.; J.H. supervised studies on CPS1 kinetics; D.M.f.v.A. supervised ETD/MS studies; N.B.-P. supervised the study, performed study concept and design, analysis and interpretation of data, and wrote the manuscript.

## Competing interests

The authors declare no competing interests. G.S.L. serves as a consultant to Audentes Therapeutics in an area unrelated to the work described in this manuscript.

## Additional information

[1]Telethon Institute of Genetics and Medicine, Pozzuoli, Italy. [2]Division of Metabolism and Children's Research Center, University Children's Hospital, Zurich, Switzerland. [3]School of Life Sciences, University of Dundee, Dundee, UK. [4]Institute of Biomolecular Chemistry, National Research Council, Pozzuoli, Italy. [5]Molecular Biology Institute, David Geffen School of Medicine at UCLA, Los Angeles, CA, USA. [6]Surgery, David Geffen School of Medicine at UCLA, Los Angeles, CA, USA. [7]Centro de Biología Molecular Severo Ochoa UAM-CSIC, CIBERER, IdiPaz, Universidad Autónoma, Madrid, Spain. [8]Department of Translational Medicine, Federico II University, Naples, Italy. [9]Scuola Superiore Meridionale (SSM, School of Advanced Studies), Genomics and Experimental Medicine Program, University of Naples Federico II, Naples, Italy. ✉e-mail: l.soria@tigem.it; brunetti@tigem.it

