## [Peer Review File · Nature Communications]

Reviewers' Comments:

Reviewer #1:

Remarks to the Author:

All questions of reviewer were properly answered by the authors and requests were met. Therefore, I suggest accepting the manuscript for publication.

Reviewer #2:

Remarks to the Author:

My critiques were addressed and this revision is improved.

Reviewer #3:

Remarks to the Author:

The revised manuscript is much improved and has answered most of my initial questions. However, there is one major concern has to be addressed.

Using genetic and pharmacological approaches, the authors showed that O-GlcNAc modified CPS1, which is associated with increased enzymatic efficiency. After mutating O-GlcNAc sites they identified, CPS1 activity was reduced, however independently of O-GlcNAcylation. Because all proteins, including wildtype CPS1 is not O-GlcNAcylated in the insect cells used. The same question was raised by Reviewer #1. Several questions and implications associated with this piece of data:

1. One could conclude from these that OGT and OGA regulate CPS1 activity independently of O-GlcNAcylation on CPS1 itself. Then what is the role of CPS1 O-GlcNAcylation? Are CPS1 O-GlcNAc levels affected by AAV-OGT or AAV-OGA in the liver?
2. It is important to study the functional outcome of CPS1 mutants. A few approaches to avoid lethality caused by CPS1 deficiency are, for example, 1) CRISPR mutagenesis, which is technically challenging; or 2) Knockdown endogenous CPS1 (e.g. targeting 3'UTR) and simultaneously express exogenous wildtype vs. mutant CPS1 in cell lines or primary hepatocytes; or 3) even simply overexpress CPS1 proteins in cells (not the insect cell line) that would O-GlcNAcylates CPS1.
3. The current Fig. 4i suggests these sites regulate CPS1 activity not through O-GlcNAcylation. Are those Thr site phosphorylated? One testable hypothesis is that OGT/OGA O-GlcNAcylate and regulate a kinase or phosphatase, that in turn modulates CPS1 phosphorylation and activity.
4. A recent publication suggests O-GlcNAcylation inhibits CPS1 activity in the liver (<https://pubmed.ncbi.nlm.nih.gov/35285892>). What are possible reasons of such discrepancy?

Response to reviewer's comments

Reviewer #3

The revised manuscript is much improved and has answered most of my initial questions.

Authors' response: We thank the reviewer for the positive feedback.

However, there is one major concern has to be addressed. Using genetic and pharmacological approaches, the authors showed that O-GlcNAc modified CPS1, which is associated with increased enzymatic efficiency. After mutating O-GlcNAc sites they identified, CPS1 activity was reduced, however independently of O-GlcNAcylation. Because all proteins, including wildtype CPS1 is not O-GlcNAcylated in the insect cells used. The same question was raised by Reviewer #1. Several questions and implications associated with this piece of data:

1. One could conclude from these that OGT and OGA regulate CPS1 activity independently of O-GlcNAcylation on CPS1 itself. Then what is the role of CPS1 O-GlcNAcylation? Are CPS1 O-GlcNAc levels affected by AAV-OGT or AAV-OGA in the liver?

Authors' response: O-GlcNAcylation of CPS1 was affected by hepatocyte-specific delivery of either OGT or OGA mediated by AAV, as shown by the newly added data (Supplementary Fig. 6c).

2. It is important to study the functional outcome of CPS1 mutants. A few approaches to avoid lethality caused by CPS1 deficiency are, for example, 1) CRISPR mutagenesis, which is technically challenging; or 2) Knockdown endogenous CPS1 (e.g. targeting 3'UTR) and simultaneously express exogenous wildtype vs. mutant CPS1 in cell lines or primary hepatocytes; or 3) even simply overexpress CPS1 proteins in cells (not the insect cell line) that would O-GlcNAcylates CPS1.

Authors' response: We performed one of the approaches suggested by the reviewer (#3) to address the functional outcome of O-GlcNAcylation on CPS1. Using a hepatocyte cell line, we showed that the mutant enzyme is less efficient at ammonia removal from media compared to the WT CPS1. This new data has been included as Supplementary Fig. 9c-d.

3. The current Fig. 4i suggests these sites regulate CPS1 activity not through O-GlcNAcylation. Are those Thr site phosphorylated? One testable hypothesis is that OGT/OGA O-GlcNAcylate and regulate a kinase or phosphatase, that in turn modulates CPS1 phosphorylation and activity.

Authors' response: the data in Fig. 4i show that independently from the post-translational modification, CPS1 enzyme activity is affected if the Thr sites are mutagenized. This finding is consistent with an important role of the Thr sites in CPS1 function, as suggested by the lack of variants in these sites in the normal population. Only one of the Thr residues was also found to be phosphorylated and we included this data in the revised manuscript. However, phosphorylation of CPS1 has not been investigated so far and kinase(s) or phosphatase(s) operating this modification are unknown. The investigation of the cross-talk between O-GlcNAcylation and phosphorylation at the Thr sites of CPS1 would be the attractive object of future studies.

4. A recent publication suggests O-GlcNAcylation inhibits CPS1 activity in the liver (<https://pubmed.ncbi.nlm.nih.gov/35285892>). What are possible reasons of such discrepancy?

Authors' response: The recent paper on CPS1 O-GlcNAcylation (Wu *et al. J Mol Cell Biol* 2022) found that CPS1 O-GlcNAcylation is increased in aged mice and reversed by caloric restriction. In contrast to our finding, this study found that O-GlcNAcylation is associated with reduced CPS1 activity. We do not know the reason for this discrepancy. However, that study was largely performed in cell culture systems and the data generated in vivo are very limited. Moreover, the in vivo studies did not include stable isotope assays to evaluate ureagenesis, a standard method to measure metabolic flux through the urea cycle. We discussed the differences between our findings and the recently published study in the discussion.

Reviewers' Comments:

Reviewer #3:

Remarks to the Author:

The revision has answered all my questions and I suggest accepting the paper for publication.

Response to the reviewer's comments

Reviewer #3 (Remarks to the Author):

The revision has answered all my questions and I suggest accepting the paper for publication.

Authors' response: We thank the reviewer for the positive feedback.